# A Review on the Interspecies Electron Transfer of Methane Production in Anaerobic Digestion System

**Kai Su *** , **Linxiao Li, Qin Wang and Rong Cao**

Faculty of Geosciences and Environmental Engineering, Southwest Jiaotong University, Chengdu 611756, China; lilinxiao@my.swjtu.edu.cn (L.L.); wq99@my.swjtu.edu.cn (Q.W.); caor@my.swjtu.edu.cn (R.C.)
* Correspondence: ksu@swjtu.edu.cn; Tel.: +86-18408256607

**Abstract:** Anaerobic methanogenesis plays an important role in the sustainable management of high concentration organic wastewater and bioenergy recovery. Interspecies electron transfer (IET) is a new type of mutualistic symbiosis that can accelerate microbial metabolism and overcome thermodynamic barriers in the metabolic process, thus facilitating anaerobic methanogenesis. IET is classified into Direct Interspecies Electron Transfer (DIET) and Mediated Interspecies Electron Transfer (MIET) according to the different electron transfer methods. This paper summarizes the recent research progress related to interspecies microbial electron transfer in anaerobic methanogenic system, describes the possible specific mechanisms of DIET and MIET, and analyzes the differences between DIET and MIET methods in terms of methanogenic performance, thermodynamics, kinetics, and the microbial communities involved in them. Finally, it was found that, through DIET, microorganisms in the process of anaerobic methanogenesis could not only strengthen the extracellular electron transfer of microorganisms and alleviate the inhibition of high organic loading rate, organic acids, and toxic substances, they could also help ferment bacteria and allow methanogenesis to break through the thermodynamic barriers and efficiently degrade complex organic matter. This can overcome several problems, such as low efficiency of electron transfer and acidification of traditional anaerobic digestion.

**Keywords:** anaerobic digestion; interspecies electron transfer; methanogenesis; acid inhibition

## 1. Introduction

Anaerobic methanogenesis is a wastewater biological treatment technology in which complex organic matter is converted to small gaseous molecules, such as $CH_4$ and $CO_2$, through an anaerobic degradation pathway that uses hydrolyzing and fermenting bacteria or hydrogen-producing acetogenic bacteria and methanogenic bacteria, which are then removed from the water [1,2]. Compared with an aerobic process, anaerobic biological treatment technology has become one of the core technologies for sustainable development of wastewater treatment because of its high treatment efficiency, energy recovery and low operation consumption [3].

The anaerobic methanogenesis process generally involves three stages: hydrolytic fermentation stage, hydrogen and acetic acid production stage, and methane production stage [4]. The anaerobic methanogenesis process is shown in Figure 1. In the hydrolysis stage, some large molecules (e.g., cellulose, starch, protein, fat, etc.) are first broken down by bacterial extracellular enzymes into small molecules (e.g., monosaccharides, disaccharides, polypeptides, long-chain fatty acids, etc.) and $H_2$ and $CO_2$ are also produced [5], and are further converted to volatile fatty acid (VFA)-based end products by fermentation bacteria. Hydrogen-producing acetogenic bacteria use the VFAs produced in the previous stage to convert fatty acids, such as propionic acid, butyric acid, and ethanol, into products that can produce methane: acetic acid, hydrogen, and carbon dioxide [6]. Methane is generally

produced by methanogenic bacteria in two ways: 1. Methane is produced from $H_2$ and $CO_2$ (Equation (1)); 2. Production of methane from acetic acid (Equation (2)) [7].

$$4H_2A \rightarrow 4A + 8H \text{ (Performed by acid-producing bacteria)}$$
$$CO_2 + 8H \rightarrow CH_4 + 2H_2O \text{ (Conducted by methanogenic bacteria)} \tag{1}$$

where $H_2A$ is the hydrogen donor.

$$C_6H_{12}O_6 + 2H_2O \rightarrow 2CH_3COOH + 2CO_2 + 4H_2$$
$$4H_2 + CO_2 \rightarrow CH_4 + 2H_2O \tag{2}$$
$$2CH_3COOH \rightarrow 2CH_4 + 2CO_2$$

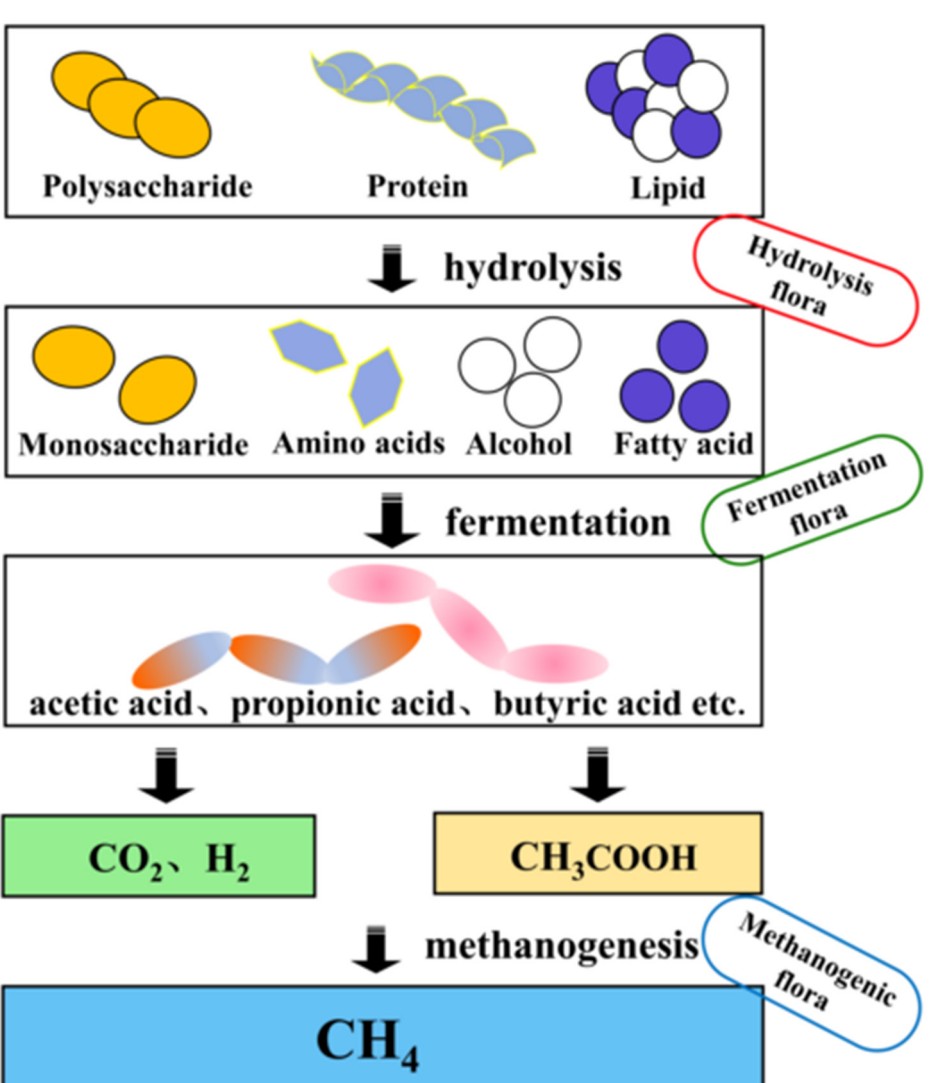

**Figure 1.** The process of anaerobic digestion.

It is generally believed that about 70% of $CH_4$ is generated from the decomposition of acetic acid during anaerobic methanogenesis, while the rest is produced from $H_2$ and $CO_2$. Studies have shown that the only substrates available to methanogenic colonies are acetic acid and one-carbon compounds, and that the proliferation and metabolic rates are slow and sensitive to environmental changes, so the methanogenic stage is considered the rate-limiting step in anaerobic digestion [8,9].

The anaerobic methanogenesis process is mediated by microorganisms of the three major bacterial groups, which form a symbiotic relationship between microorganisms, thus overcoming the thermodynamic barriers of the metabolic process. In symbiotic

relationships, interspecies electron transfer (IET) is a new type of mutualistic symbiosis that has been discovered in recent years. Electron donor microorganisms transfer electrons to electron acceptor microorganisms by direct means of cell contact or indirect pathways mediated by intermediates, thus enabling metabolic processes that are difficult for a single microorganism to accomplish. IET can be divided into Direct Interspecies Electron Transfer (DIET) and Indirect Interspecies Electron Transfer (MIET) according to the different modes of electron transfer [10].

In the formation of methane via hydrogen, hydrogen can be considered a diffusive electron carrier, and this process is thermodynamically feasible, i.e., $\Delta G < 0$, only if the hydrogen produced by the acetic-acid-producing bacteria is efficiently used by the hydrogen-consuming methanogenic bacteria, and if the hydrogen in the system maintains a very low partial pressure ($H_2 < 10^{-4}$ atm) [11]. Formic acid and electron transporters also play a similar role to hydrogen in the formation of methane [12,13]. However, the efficiency of MIET is relatively low due to the diffusion limitation of the electron carriers, so IET via hydrogen, formic acid, and electron-delivered substances tends to be the limiting step in the methanogenesis process and is considered to be the bottleneck of methane formation [14,15]. It has also been reported that DIET can be performed through cytochromes, conductive pili, and cellular appendages on the cell membrane, as well as some conductive materials, and that the methanogenic process through DIET is more efficient in electron transfer compared to MIET [16,17]. Most studies on DIET have shown a shortened lag time for methanogenic processes, increased methane production, and resistance to inhibitory conditions. Therefore, DIET can be an effective alternative to IHT (Interspecies Hydrogen Transfer)/IFT (Interspecies Formic Transfer) [18].

Despite the increasing number of studies on DIET, most of the current literature focuses on batch and continuous flow experimental manipulations with different inoculums and substrates as well as their mechanistic analysis. Therefore, based on the review of the above-mentioned articles, this paper summarizes the research on the mode and mechanism of electron transfer through IET, the comparison of two modes of IET and the progress of research on direct interspecies electron transfer to alleviate the inhibition effect of anaerobic digestion. By reviewing the above, we hope to show the direction for the potential industrial applications of DIET. Finally, the challenges and prospects for the development of DIET in anaerobic digestion are discussed in the hope of providing new ideas for the research direction in this field.

## 2. The Modes and Mechanisms of Electron Transfer

Electrons are one of the fundamental particles of matter. The gain or loss of electrons is accompanied by the breakage and formation of chemical bonds, the flow of electrons drives the synthesis and release of energy, and electron transfer is the basis of all life activities in living systems [19]. Interspecies electron transfer is an important element of material and information exchange between microbial populations and is key to building interspecies mutualistic relationships among microbial populations. For anaerobic methanogenesis, interspecies microbial electron transfer has been considered a key link between the acid-producing fermentation stage and the anaerobic methanogenesis stage [20,21] because there is a rate-limiting process in the transition between acid production and methane production processes in anaerobic digestion.

### 2.1. MIET

2.1.1. Hydrogen-Mediated MIET

Interspecies Hydrogen Transfer (IHT) has been considered a major discovery in anaerobic methanogenesis. IHT is a theoretical basis for the mutual metabolism of two groups of microorganisms, acetic acid-producing bacteria and methanogenic archaea, providing technical support for the stable operation of anaerobic biological treatment processes [20,22]. In 1967, Bryant et al. discovered that the ethanolic methanogenic "*Methanobacillus omelianskii*" is in fact composed of two intercalating bacteria, in which the S strain oxidizes ethanol

to $H_2$ and acetic acid, and the M.o.H strain uses $H_2$ and $CO_2$ to produce methane, which is metabolized by IHT The reciprocal metabolism is carried out by IHT [23]. This study shows that $H_2$ is an intermediate carrier for electron transfer between these two bacteria in methanogenesis process, tentatively confirming the existence of MIET. The specific responses are as follows.

S strain:

$$CH_3CH_2OH + H_2O \rightarrow CH_3COO^- + H^+ + 2H_2 \ \Delta G^0 = +9.5 \ kJ \cdot mol^{-1} \quad (3)$$

M.o.H strain:

$$CO_2 + 4H_2 \rightarrow CH_4 + 2H_2O \ \Delta G^0 = -131 \ kJ \cdot mol^{-1} \quad (4)$$

Co-cultivation system:

$$2CH_3CH_2OH + CO_2 \rightarrow 2CH_3COO^- + 2H^+ + CH_4 \ \Delta G^0 = -112 \ kJ \cdot mol^{-1} \quad (5)$$

From Equation (3), it is clear that the process is energy consuming and the only way to carry out the acetic acid production reaction is to reduce the hydrogen partial pressure. IHT has been considered to be the rate-limiting step of anaerobic methanogenesis process due to the positive Gibbs free energy of the conversion reaction of VFAs, such as butyric acid and propionic acid, under standard conditions. This, along with the extremely low solubility of $H_2$ in water, makes it difficult to proceed spontaneously. At the same time, the $H_2$ reduction reaction of IHT requires the participation of various enzymes. Since this makes it easy for energy loss to occur, the reaction is not conducive to microbial growth [24].

### 2.1.2. MIET Mediated by Formic Acid

Formic acid can also act as an electron carrier to mediate the occurrence of MIET; this process is known as Interspecific Formic Transfer (IFT) [25]. In 1988, Thiele et al. found little metabolite $H_2$ in a methanogenic reactor constructed by *Desulfovibrio vulgaris* and *Methanobacterium formicicum*, and the addition of $H_2$ did not significantly promote methane production, which revealed that the reaction system relied on formic acid for mediation [12]. In IFT, formate dehydrogenase couples oxidation with the electrons obtained from the substrate to reduce $CO_2$ to formate. Formic acid is involved in methanogenesis through two pathways: cleavage to $H_2$ and $HCO_3^-$ (or $CO_2$) to produce $CH_4$ directly or oxidation by formic acid dehydrogenase to produce methane [26]. However, the activity of formate dehydrogenase in some colonies is very low, so IHT is considered the most typical mode of electron transfer in methanogenic mutualistic metabolism [27].

### 2.1.3. E-Transmitter Mediated MIET

In addition to intermediate metabolites, electron transmitter with redox properties can also mediate microbial MIET. Depending on the source, electron transmitter can be divided into two types: small molecules secreted by the cells themselves, such as riboflavins, phenazines and quinones; and natural or synthetic compounds, such as humic substances and neutral reds [28]. Huang et al. found that riboflavin-mediated MIET was present in *G. metallireducens* and *G. sulfurreducens* co-culture systems [29]. Among them riboflavin promotes MIET between free state bacteria in the form of electron transmitter [29]. The possible mediated interspecies electron transfer mechanisms are shown in Figure 2.

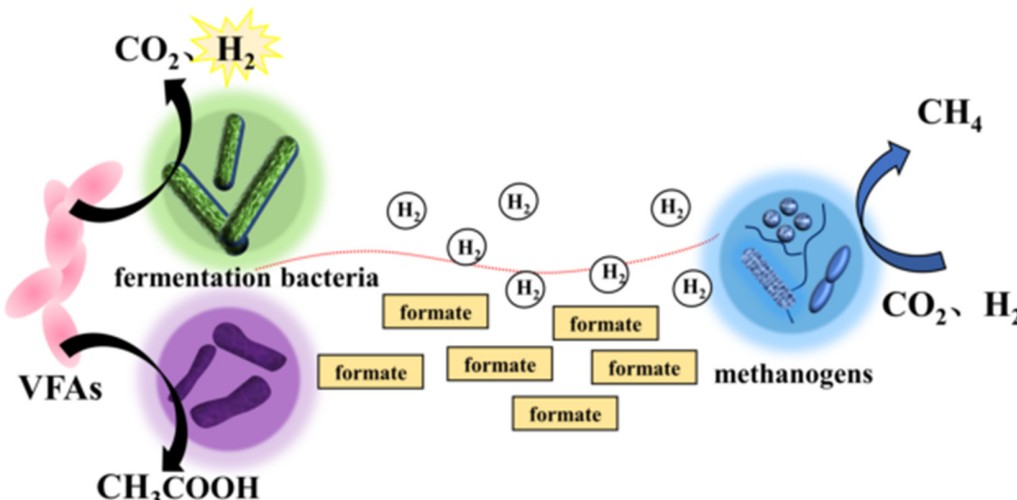

**Figure 2.** Possible mediated interspecies electron transfer mechanism between fermentation bacteria and methanogens via hydrogen/formate.

### 2.2. DIET

2.2.1. DIET via Bioelectric Connection

In recent years, studies have reported DIET with higher interspecies electron transfer efficiency, where certain bacteria can transfer electrons directly to methanogenic bacteria through direct interbacterial contact [30]. DIET by bioelectric linkage is a way to directly exchange interspecies electrons by forming tight junctions on the outer surface of cells using their own cellular structures, such as conductive pili and cytochrome c [31]. In 2005, Reguera et al. used conductive probe atomic demonstrated the high electrical conductivity of bacterial pili, hair-like conductive appendages that grow on the surface of bacteria and exhibit metal-like conductivity [32,33]. Summers et al. 2010, in a study of co-culture systems of *G. metallireducens* and *G. sulfurreducens*, found that the *G. sulfurreducens* hydrogenase knockout strain was able to co-culture with *G. metallireducens* despite its inability to utilize $H_2$; however, when the Multi-haem cytochrome genes *omcS* and the gene *pilA* related to *pili* synthesis in *G. sulfurreducens* were knocked out in the co-culture system, the growth of the bacterial was found to be inhibited [18]. The results of this experiment suggest that electron transfer in the co-culture system is carried out via conductive bacterial pili-linked DIET. Immediately after, in 2011, Morita et al. similarly observed that conductive pili mediated DIET in an upflow anaerobic sludge reactor for the treatment of beer wastewater [34]. Subsequently, in a study by Rotaru et al. in 2014, *G. metallireducens* was found to produce methane by DIET co-culture with *Methanosaeta harundinacea* or *Methanosarcina barkeri* via conductive pili that using ethanol as a substrate [35,36].

In addition to pili that can mediate DIET, multi-haem cytochromes can also exchange electrons, thus completing interspecies electron transfer [37]. It has been shown that DIET can also be formed between *G. sulfurreducens Aro-5* strains with poorly conductive pili and *G. metallireducens*. Ueki et al. and Liu et al. found that two genus Geobacter without pili can also grow by forming aggregated granules through DIET, and that the key role is played by the cytochrome encoded by *Gmet*-2896 in *G. metallireducens* [38,39]. According to a study by Lovley in 2017, when close contact junctions are formed between microbial cells, the use of conductive pili as a means of DIET mediation becomes less important, and *Prosthecochloris aestuarii* can absorb the *G. sulfurreducens* release directly through close contact, without passing through conductive pili electrons [10]. A similar phenomenon was previously found by McGlynn et al. in a co-culture of methanogenic and sulfate-reducing bacteria [37].

2.2.2. DIET Connected by Conductive Material

In addition to the examples mentioned above, with DIET mediated by the cell's own structure, researchers discovered in 2012 that the addition of conductive materials during

anaerobic methanation can also contribute to DIET [14]. Generally, conductive materials are classified into carbon-based and iron-based conductive materials according to their properties. The most widely studied carbon-based materials include biochar, activated carbon, graphene, and carbon cloth [40]; Iron-based conductive materials include magnetite, iron oxide, hematite, red mud, etc. [40]. The results of numerous studies have shown that these materials can act as conductors to promote DIET and improve electron transfer efficiency, resulting in a faster rate of anaerobic methanogenic process and higher methane yield [41].

Rotaru et al. used ethanol as a metabolic substrate and found that modified granular-activated carbon could replace conductive bacterial pili for electron transfer in the co-culture system of *Geobacter metallireducens* and *Methanosarcina barkeri*, and that it has better methanogenic performance than the bacterial pili [35]. Not coincidentally, Chen et al. in 2014 also found that the addition of biochar to the co-culture system of *G. metallireducens* and *G. sulfurreducens* and the co-culture system of *G. metallireducens* and *Methanosarcina barkeri* also significantly promoted methane production [42]. In 2015, Luo et al. found that the addition of biochar shortened the anaerobic reaction lag period while resulting in a significant increase in methane production through batch experiments [43]. In the same year, Zhao et al. studied the effect of graphite column, biochar, and charcoal cloth on the treatment effect of anaerobic bioreactor using ethanol as carbon source, and found that the experimental group with the addition of carbon-based conductive material had different degrees of improvement in COD removal and methane production compared with the control group, and the reinforcement effect of charcoal cloth was better than that of graphite column and biochar [44].

Kato et al. demonstrated that the addition of iron oxide nanoparticles (10–50 nm), such as magnetite or hematite, to paddy soil contributed to the enrichment of methanogenic microorganisms, enhanced interspecies interactions, and promoted methanogenesis [45]. Liu et al. reported that the addition of magnetite to a *Geobacter* co-culture system containing *OmcS* deletion mutants resulted in the re-formation of DIET. In addition, the presence of magnetite decreased the expression of the *omcS* gene in the wild-type *Geobacter* co-culture system [46]. Meanwhile, a study by Wang et al. found that the addition of magnetite during digestion of high-solids sewage sludge alleviated the accumulation of short-chain fatty acids, accelerated methanogenesis, and reduced the expression of pili and c-type cytochromes, suggesting that magnetite could be used for extracellular electron transfer by replacing c-type cytochromes [47]. Typical conductive materials involved in DIET are shown in Table 1. The possible direct interspecies electron transfer mechanisms are shown in the Figure 3.

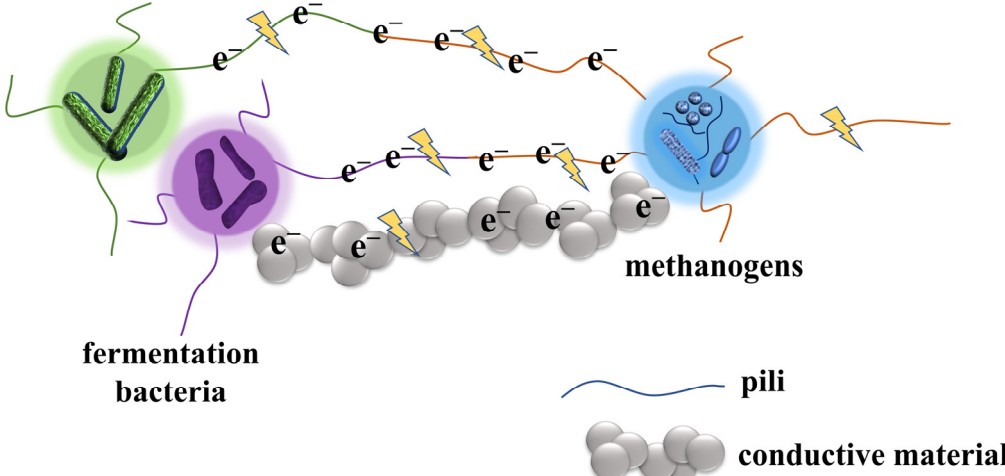

**Figure 3.** Possible direct interspecies electron transfer mechanism between fermentation bacteria and methanogens via pili/conductive material.

**Table 1.** Typical conductive materials involved in DIET.

| Conductive Materials | Dose | Operation Mode | Substrate | Main Effects and Impacts of Promotion | References |
|---|---|---|---|---|---|
| Biochar | 1.25 g/L | Reactor | Ethanol | CH4 production rate increased by 30–45%. | [44] |
| Granular activated carbon | 10 g/L | Bacth | Glucose | CH4 production rate increased by 168%; Accelerated substrate hydrolysis. | [48] |
| Graphene | 1.0 g/L | Bacth | Ethanol | Accelerated hydrolysis and acidification of substrates. | [49] |
| Carbon Cloth | 10 g/L | Bacth | Ethanol | Accelerated hydrolysis and acidification of substrates. | [42] |
| NZVI | 20 mg/L | Reactor | Pig manure | Adding 20 mg/L NZVI increased CH4 by up to 126% for digesting of pig manure. | [50] |
| | 0.1% wet wight of sludge | Bacth | Sludge | Accelerated hydrolysis and acidification of substrates. | [51] |
| Magnetite | 25 mM | Bacth | Acetate | Magnetite supplementation accelerated thermophilic methanogenesis; CH4 production rate increased by 130%. | [52,53] |
| Iron Oxide | 750 mg/L | Reactor | Beet sugar industrial wastewater | Accelerated hydrolysis process of substrates. | [54] |
| Red mud | 20 g/L | Bacth | Waste activated sludge | 136% increase in methane production compared to the control group. | [55] |

## 3. Comparison of DIET and MIET

### 3.1. Effect of DIET and MIET on Methanogenic Performance

Electron transfer by MIET often has the disadvantages of electron carrier limitation and is susceptible to external environmental influences, such ase.g., organic load, fatty acid concentration, and $H_2$ partial pressure, which often results in a low efficiency of anaerobic methanogenesis. Co-metabolism in MIET occurs mainly through IHT. The IHT process requires the synergistic action of hydrolytic fermentation bacteria and methanogenic bacteria, and the acetic acid production reaction can only occur spontaneously under conditions of very low hydrogen partial pressure. The $H_2$ reduction reaction of IHT requires the participation of various enzymes, which also easily causes energy loss and is not conducive to microbial growth [24]. When $H_2$ is over-produced and under-consumed, the partial pressure of $H_2$ in the system will continue to rise, leading to the accumulation of VFAs and a serious stagnation of the anaerobic methanogenic system [20]. However, in methanogenesis mediated by DIET, electrons are not transferred by diffusive electron carriers, but instead by either direct electron transfer through their own conductive cellular structures or an additional conductive material [56]. The increased efficiency of electron transfer is reflected in the increased rate of the methanogenesis process.

Wang et al. showed that the addition of ZVI could effectively promote the production of medium-chain fatty acids (MCFAs) in waste activated sludge (WAS) and enhance the degradation of WAS in anaerobic WAS fermentation. The maximum yield of MCFAs was 15.4 g COD/L, and the selectivity of MCFAs was 71.7% at 20 g/L of ZVI, which were 5.3 and 4.8 times higher (2.9 g COD/L and 14.9%) than the control, respectively. This is because ZVI

reduces the redox potential (ORP) and creates a more favorable environment for anaerobic biological processes; ZVI with strong electrical conductivity helps to improve the electron transfer efficiency from electron donor to acceptor and promotes DIET [57]. According to Yan's study, ZVI may act as an intracellular electron shuttle effectively facilitating the metabolism of propionate and butyrate as well as DIET-mediated methanogenesis [58]. In addition to ZVI, Wang et al. examined the enhanced effect of magnetite on anaerobic methanogenesis and found that the addition of 50 mg/g TS magnetite increased the rate of methanogenesis by 26.6%, that magnetite promoted DIET, and that the enhanced DIET partially replaced IHT; they also increased the acetic acid-based methanogenic pathway. Li et al. explored the effect of $nFe_3O_4$ on anaerobic digestion by injecting nanometer triiron tetroxide ($nFe_3O_4$) into an anaerobic reactor. The results showed that the addition of $nFe_3O_4$ increased the methane production and COD removal of the reactor by 403.7% and 33.1%, respectively. The addition of the conductive material $nFe_3O_4$ stimulated the secretion of a large amount of electroactive material, which acted as an electron shuttle and promoted the DIET [31].

Iron-based conductive materials, such as magnetite, ZVI, and stainless steel, can increase methane production by 1.1–4.4 times and methane yield by 15–240%. Meanwhile, carbon-based materials, such as GAC, biochar, carbon nanotubes, and graphene, can increase methane production by 13–140% and methane yield by 20–570% [40]. Carbon materials can enhance the anaerobic methanogenesis process by promoting microbial growth and aggregation, increasing enzyme activity and buffering capacity, accelerating IET, and promoting the formation and utilization of VFAs [59]. Hao et al. investigated the effect of multi-walled carbon nanotubes on methane production in an anaerobic methanogenic system and showed that the presence of 500 mg/kg multi-walled carbon nanotubes increased the cumulative daily methane production by about 46.8%, reduced the total solids content by 12.8% and also reduced the pH, and significantly increased the relative abundance of methanogenic bacteria [60]. In addition, carbon nanotubes as conductive materials can enhance the DIET between syntrophic acetogenic bacteria and methanogenic bacteria. Casting carbon nanotubes into the anaerobic digestion system can not only provide an electronic conductor for the IET process of anaerobic microorganisms, but can also ultimately promote the methanogenic process by increasing the content of EPS and the relative abundance of methanogenic bacteria [48].

### 3.2. Comparison of Degradation Process from Thermodynamic and Kinetic Perspective

The main limitation in the MIET process is that any stagnation of the process will lead to the accumulation of VFAs; propionic acid is the VFA that has the most significant toxic effect on microorganisms [61]. From a thermodynamic point of view, DIET degrades propionic acid more energy efficiently than MIET. This is because the generation and consumption of electron carriers in MIET requires multiple enzymatic reactions, each requiring energy consumption [62]; DIET, meanwhile, does not require complex enzymatic reactions to generate electron carriers, consumption of $H_2$, or diffusion of redox mediators. The study by Jing et al. enumerated that the oxidation of propionic acid by propionic acid oxidizing bacteria [63]. The standard Gibbs free energy of MIET and DIET are shown in the following equations, respectively.

$$C_2H_5COO^- + 3H_2O \rightarrow CH_3COO^- + HCO_3^- + H^+ + 3H_2 \qquad (6)$$

$\Delta G$ = 72.7 kJ/mol

$$C_2H_5COO^- + 0.75H_2O \rightarrow CH_3COO^- + 0.25HCO_3^- + 0.25H^+ + 0.75CH_4 \qquad (7)$$

$\Delta G$ = −26.4 kJ/mol

From the above equation, it can be seen that DIET is more energy efficient compared to MIET. Therefore, DIET-mediated methanogenesis does not need to be influenced by the

electron carrier or electron delivery concentration and, thermodynamically, would be more likely to occur.

Numerous studies since 2012 have compared methane production and generation rates in anaerobic digestion systems with and without the use of conductive materials. They have demonstrated that DIET can enhance the reaction kinetics in anaerobic methanogenic processes [64]. The addition of conductive materials differentially enhances the kinetic characteristics of methanogenesis, and the potential reason for promoting the methanation process may be the promotion of DIET reaction kinetics by iron oxides. Cruz Viggi et al. showed that the addition of magnets made the reactor current intensity 106 times higher than that of the blank reactor [17]. Yin et al. found that the electron transfer coefficient Kapp was 84.2% higher in the reactor with the addition of $Fe_3O_4$ than in the blank reactor. In addition, the study used a cyclic voltammetry experiment and calculated electron transfer-related quantities, such as electrical conductivity based on relevant equations, and these data indicated that $Fe_3O_4$ accelerated the rate of extracellular electron transfer [65]. According to the dynamic study of VFAs by Yin et al. in 2017, the addition of $Fe_3O_4$ accelerated the consumption of VFAs [66] and the hydrolysis and acidification of high tryptophan content [67]. The results of the above studies directly indicate the degree of enhancement of DIET reaction kinetics from the electron transfer efficiency or current magnitude.

DIET has certain advantages over MIET in terms of both thermodynamics and kinetics, but it can only accomplish electron transfer between hydrolytic acidifying bacteria and methanogenic bacteria through direct contact with conductive pili, cytochromes, or conductive materials, which is limited in spatial distance. Compared to DIET, MIET does not require direct contact and can achieve longer distance electron transfer through electronic carriers. The comparison of MIET and DIET is shown in Table 2.

**Table 2.** Comparison of MIET and DIET.

| | MIET | DIET | References |
|---|---|---|---|
| Mechanism | Mutual symbiosis using electronic carriers (Electronic carriers: Hydrogen, Formic acid, L-Cysteine, Sulfide, Quinones, Riboflavin, Phenazine) | Mutual symbiosis using direct contact of the bacterium (Self-structure of bacteria: Conductive pili, Cyt-c; External conductive material: Activated Carbon, Magnetite, etc.) | [12,29,68–71] |
| Advantages | Longer distance electron transfer is possible | 1. High efficiency of electron transfer 2. No need to complete complex enzymatic reactions, saving energy 3. No limitation of electron mediator type and diffusion efficiency | [28,35,36,72] |
| Limitations | 1. Limitation of hydrogen partial pressure 2. MIET requires the synthesis of a variety of enzymes, which can easily cause energy loss | 1. Limited in terms of spatial distance 2. Limited by microbial species and the activity of redox proteins | [20,73] |

### 3.3. Differences in the Microbial Communities Involved in MIET and DIET

In the microbial IET process, microorganisms can be classified into electron donor microorganisms and electron acceptor microorganisms according to the different electron-producing and accepting subjects. Most syntrophic acetogenic bacteria can perform hydrogen- or formic acid-mediated MIET. The more studied electron donor microorganisms are the *Syntrophobacter* and *Desulfovibrio*: *D. vulgaris* [74], *Syntrophomonas wolfei* [75], *P. carbinolicus* [76], *Pelotomaculum* [77], *Desulfotomaculum* [78]; electron acceptor microorganisms (methanogenic archaea) contain *Methanococcales*, *Methanobacteriales*, *Methanomicrobiales* and *Methanopyrales*. Compared to MIET, the species of microorganisms that can be subjected to DIET are more limited. The electron donor bacteria for DIET are mainly electron-producing microorganisms, such as

*Geobacter*, which has strong electron-producing ability and can transfer electrons across the membrane to the extracellular area; athte same time, it can synthesize conductive bacterial pili or cytochromes. The presence of conductive pili and cytochromes is necessary for the formation of DIET [38,79]. Donor microorganisms include *G. mentallireducens* GS-15 [18], *G. sulfurreducens* PCA, *G.hydrogenophilus* [80] and anaerobic methanotrophic [37,81] etc. The electron acceptor microorganisms involved in DIET are *G. sulfurreducens* PCA [18], *M.barkeri* [35,36], *Methanosarcina mazei* [82], *M.harundinacea* [35,36], *Methanobacterium* sp. YSL [83], *Thiobacillusdenitrificans* [45], *P. aestuarii* [84] and *R. palustris* [85] etc. The methanogenic bacteria with DIET ability are mainly distributed in acetotrophic methanogens, including *Methanothrix* and *Methanosarcina*, such as *Mx. harundinacea*, *M. mazei*, *M. acetivorans*, *M. horonobensis*, *M. barkeri*, *Methanosarcina vacuolata*, etc. However, most methanogenic bacteria are unable to synthesize conductive hairs or cytochromes, but can accept electrons by contact with solid electrodes or by DIET.

MIET and DIET are not independent modes of electron transport in methanogenic intermicrobial flora. For example, bacteria that typically rely on MIET to syntrophy with methanogenic archaea, such as *Syntrophus acidtrophius* [86], also have the ability to DIET because of the electrical conductivity of their pili [87]. *Desulfovibrio*, on the other hand, has been studied as a model strain for interspecific hydrogen or formic acid transfer, but a study by Zheng et al. in 2021 suggests that it may have recently been found to possess DIET [88]. Similarly, studies by Holmes, Liu, and Rotaru et al. demonstrated that, for methanogenic bacteria, *M. barkeri* can undergo both hydrogen-dependent MIET with *Pelobacter* [76]. DIET can also occur with *G. metallireducens* [35,85]. Therefore, the IET pattern of microorganisms in nature may be the coexistence of multiple IET modes, closely related to a variety of conditions, such as environmental conditions, and other microbial classes. Some of the microorganisms involved in IET and their IET patterns are shown in Table 3.

**Table 3.** Some of the microbial species involved in IET and their IET pattern.

| Electron-Donating Microorganism | Electron-Accepting Microorganism | IET Pattern | References |
|---|---|---|---|
| *S* strain | *M. ruminantium* | MIET (H$_2$-mediated) | [23] |
| *D. acatoxidans* | *P. aestuarii* | MIET (Sulfide-mediated) | [69] |
| *G. sulfurreducens* | *W. succinogenes* | MIET (L-cystine/cysteine-mediated) | [68] |
| *D. vulgaris* | *Methanobacterium formicicum* | MIET (Formate-mediated) | [74] |
| *Syntrophomonas wolfei* | *M. barkeri* | MIET (Formate-mediated) | [75] |
| *Pelotomaculum* | *Methanobacteriaceae* | MIET (Cysteine-mediated) | [77] |
| *G. metallireducens* | *G. sulfurreducens* | DIET | [18] |
| *Geobacteraceae* | *M. mazei* | DIET | [82] |
| *G. metallireducens* | *Methanobacterium* sp. YSL | DIET | [83] |
| *Desulfovibrio* sp. | *Methanobacterium electrotrophus* | DIET | [88] |
| *Rhodoferrax ferrireducens* | *Mx. harundinacea* | DIET | [89] |
| *G. hydrogenophilus* | *M. barkeri* | DIET | [80] |

## 4. DIET Alleviates Inhibition in Anaerobic Methanogenesis

### 4.1. Mitigation of the Inhibitory Effect of DIET on High Organic Loading Rate

One of the most critical process parameters in conventional anaerobic methanogenesis is the organic loading rate (*OLR*), which determines the methanogenic activity and methanogenic kinetics to a certain extent [90]. Hydrogen-, formic acid-, or redox-mediated MIET can all cause stagnation in anaerobic digestion and problems leading to accumulation of VFAs, and high concentrations of VFAs (especially propionic acid) are toxic to methanogenic bacteria [91]. In addition to the problem of VFAs' accumulation. This type of MIET also leads to the accumulation of $H_2$ and the decrease of pH.

Adding conductive materials, such as carbon cloth, GAC, magnetite, and biochar, to the reactor with high *OLR* can promote the establishment of DIET between electricity-producing bacteria and methanogenic archaea to form efficient electron transfer channels, enhance the synergistic metabolism among microorganisms, promote the conversion of organic matter and alleviate the accumulation of VFAs, thus promoting the methane production and methanogenic rate of anaerobic systems. A study by Zhao et al. proposed the use of ethanol as a DIET substrate culture, which was used to stimulate the anabolism of propionic acid and butyric acid in the microbial community. The results showed that synthetic degradation of propionic or butyric acids in the reactor was significantly improved when propionic acid or butyric acid was used as the sole carbon source, and according, to microbial community analysis, *Geobacter* that performed DIET were only detected in the reactor with ethanol added. With a significant increase in the number of *Methanosaeta* and *Methanosarcina*, a potential DIET between *Geobacter* and *Methanosaeta* or *Methanosarcina* may be established, which then exhibits an accelerated synthetic degradation of propionic and/or butyric acid in the system. Their further experiments showed that granular activated carbon (GAC) could also promote the anabolism of propionic and butyric acids in an ethanol-stimulated enrichment [92]. Jing's experimental results showed that the addition of 10 mg/L conductive magnetite increased the propionic acid-methanation yield by about 44%, and the enrichment of *Thauera* after the addition of magnetite may be related to DIET. According to the result table of iTRAQ quantitative proteomics analysis and the determination of cytochrome *c* oxidase, magnetite promotes DIET and IHT during propionic acid-methanation [63]. In summary, DIET can alleviate the inhibitory effect of high *OLR* on anaerobic methanogenesis.

### 4.2. Mitigation of Organic Acid Inhibition by DIET

The biggest problem in operating a reactor under high organic loading conditions is acidification [93]. The rate of fermentation in an anaerobic system is greater than the rate of methanogenesis at increased substrate concentrations because fermenting bacteria grows more quickly than methanogenic bacteria [94]. The rapid growth of acid-producing bacteria leads to an excessive accumulation of VFAs, resulting in the accumulation of short-chain fatty acids and alcohols produced by fermentation in the reactor, slowing or even stalling the anaerobic methanogenesis rate, in which case the pH in the reactor drops rapidly and irreversible acidification occurs [95,96]. Since methanogenic bacteria are a class of strictly anaerobic bacteria that are sensitive to environmental changes and extremely demanding, a decrease in pH will lead to a decrease in the activity of methanogenic bacteria due to the limitation of their growth and metabolism [97]. The bacterial population in the reactor is out of balance and eventually destabilizes and collapses.

Wang et al. found that the addition of biochar to the mesophilic anaerobic reactor was found to significantly increase the methane yield as well as reduce the lag time; they also found that the addition of biochar could alleviate the acidification phenomenon due to the accumulation of VFAs, as the concentration of oxidized VFA in the DIET reactor was 63.5% of the control value, and the maximum VFA consumption rate was 57.5%. The rapid consumption of VFA increased methane production by 27%, and this study suggests that biochar mediates DIET between electron donor and electron acceptor microorganisms [47]. Previous research by Dang et al. suggested that fermenting bacteria may alter the metabolic pathway of complex

organic matter conversion to acetate and $CO_2$ directly, while also transferring electrons directly to methanogenic bacteria via DIET. This may reduce the production of butyric acid, propionic acid, and $H_2$ [98]. Cruz et al. investigated the effect of adding micron-sized magnetite on the anaerobic digestion of propionate by using Confocal Laser Scanning Microscopy (CLSM) and Fluorescence In Situ Hybridization (FISH), which showed that aggregating microorganisms and archaea were always close to bacteria, and thus hypothesized that the addition of magnetite promoted the establishment of DIET mechanism and enhanced the degradation efficiency of propionate [17]. Both Zhuang et al. and Baek et al. demonstrated that DIET by adding conductive materials promoted the consumption of organic acids and increased the efficiency of the conversion of organic acids to methane, and that the cultured microorganisms were better adapted to the stress of high concentrations of VFAs [99,100]. In summary, DIET can effectively alleviate acid inhibition under high loading organic conditions in conventional anaerobic methanogenesis and can avoid the problems of low methane yield due to high $H_2$ partial pressure in MIET-mediated methanogenesis.

*4.3. Mitigation of Toxicant Inhibition by DIET*

Aromatic compounds are a class of persistent, bioaccumulative, and toxic organic pollutants. Anaerobic microorganisms are more sensitive to the environment, and high concentrations of aromatic compound wastewater can have a significant inhibitory effect on their activity. Aromatic compounds are a class of compounds that have an aromatic ring structure. When hydrolytic acidifying bacteria and methanogenic archaea are inhibited by their toxicity during anaerobic methanation, the methanogenic process is slowed down or even stalled. DIET promotes the metabolic activity of mutualistic microorganisms, thus enhancing the catabolic utilization of aromatic compounds. After adding bio-nano-FeS or magnetic carbon to the anaerobic reactor, Li et al. found that the addition of bio-nano-FeS or magnetic carbon could improve the efficiency of methane production and pyrene degradation by promoting DIET between bacteria and methanogenic bacteria. The researchers added 200 mg/L of bio-nano-FeS or magnetic carbon to the anaerobic digestion process for 25 days and found that the system reduced short-chain fatty acids, increased methane production by 58.1% and 33.4%, respectively, and achieved 77.5% and 72.1% removal of pyrene, compared to 40.8% in the control system [101]. Zhuang et al. established DIET between mutualistic benzoate oxidizing and sulfate-reducing bacteria promoted by the addition of hematite and magnetite in an anaerobic digestion reactor, which increased the degradation of benzoate by 81.8% and 91.5%, respectively [102].

During anaerobic methanogenesis, some organic nitrogen, such as protein, urea and nucleic acid, will be converted to inorganic ammonia nitrogen by microbial action. However, as the rate of ammonia nitrogen production is greater than its utilization rate, it will accumulate and inhibit anaerobic microorganisms, slowing anaerobic methanogenesis [103]. Zhuang et al. showed that, when the ammonia nitrogen concentration reached 1500–1700 mg/L, it would strongly inhibit the anaerobic methanogenesis process [99]. The high concentration of ammonia nitrogen inhibits the acetic acid production process, which leads to the accumulation of VFAs; combined with the fact that methanogenic bacteria are more sensitive to ammonia nitrogen than hydrolytic acidifying bacteria, this further reduces the activity of methanogenic bacteria [104–106]. In recent years, the relationship between DIET and anaerobic digestion, mitigation of ammonia nitrogen inhibition has been studied by many researchers, and preliminary results have been obtained. In 2018, Zhuang et al. found that the methane yield in the reactor with magnetite addition was enhanced by 58% compared to the control at a high ammonia nitrogen concentration of 5 g/L. This was due to the addition of magnetite that may have promoted DIET between acetate oxidizing bacteria and hydrotropic methanogenic archaea [99]. In 2019, Lee et al. also found that the addition of magnetite reduced the delay period of anaerobic digestion by 21% in the case of 6.5 g/L ammonia nitrogen inhibition [107]. In the same year, Lu et al. suggested that the addition of magnetite could promote DIET between hydrotropic methanogenic bacteria and acid-producing bacteria at high ammonia nitrogen concentrations, thus promoting

anaerobic methanogenic processes [108]. Therefore, the establishment of DIET makes it possible to alleviate the inhibition of ammonia nitrogen and enhance the reaction efficiency.

DIET can also alleviate the inhibition of methanogenic processes caused by high sulfide concentrations during anaerobic methanogenesis. Jin et al. added magnetite to sulfate-containing wastewater to increase its methane production by 3–10 times; it was found that the concentration of c-type cytochrome in the experimental group with magnetite addition was 113.54 nmol/L, which was about half of that in the control group. Fe(III)-reducing bacteria *Veillonella* were found to be enriched on the surface of magnetite and were associated with methanogenic bacteria *Methanothrix* and *Methanosarcina* established DIET [109]. A study by Li et al. examined the addition of stainless steel to sulfate-containing wastewater and found that the average methane production of the reactor with stainless steel strips was 4.5 times higher than that of the control. The kinetic advantage of DIET for electron donors was also found to be 108 times higher than that of IHT [110]. According to the above study, the establishment of DIET can alleviate some of the problems brought about by conventional MIET, such as inhibition of methanogenic processes in reactors under high loading conditions and inhibition of organic acid accumulation and toxic substances. Alleviation of inhibition of anaerobic methanogenesis by DIET is shown in the Figure 4.

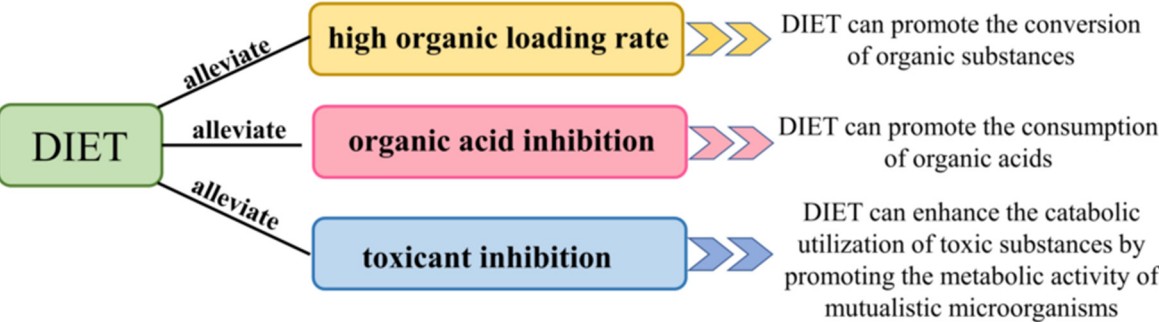

**Figure 4.** DIET alleviates inhibition in anaerobic methanogenesis.

### 5. Conclusions and Prospects

In recent years, considerable progress has been made into research on IET, but research on DIET is still in the initial stage. On this basis, this paper reviews the research on IET in recent years and describes the mechanism and significance of the two main modes of IET, MIET and DIET. This paper compared the advantages and disadvantages of MIET and DIET in the process of anaerobic methanogenesis and from the viewpoint of thermodynamics and kinetics and found that DIET not only enhances the extracellular electron transfer of microorganisms during anaerobic methanogenesis, it also helps fermenting bacteria and methanogenesis to break through the thermodynamic barriers and efficiently degrade complex organic matter, overcoming the difficult problems, such as low efficiency of electron transfer and acidification, in traditional anaerobic digestion.

Anaerobic methanogenesis plays an important role in the sustainable management of high concentration organic wastewater and bioenergy recovery. The discovery of DIET between electroactive microorganisms and anaerobic methanogenic bacteria provides a new approach and idea for solving the problem. Follow-up research can be conducted in the following areas:

1. Discovery of more anaerobic methanogenic bacteria capable of direct interspecies electron transfer with electron-producing microorganisms. With the development of bioelectrochemistry, electron-producing microorganisms and their pathways of electron transfer have received extensive attention. However, a limited number of microorganisms capable of driving IET have been identified. Among the methanogenic groups, only *Methanosarcinales* and *Methanobacterium* have been found to be capable of DIET with electrogenic microorganisms; however, there are a large number of unexplored microorganisms involved in the DIET process. For example, *Methanomicrobia*,

*Methanobacter*, *Methanolinea*, and *Methanospirillum* were found to have potential DIET capabilities in a study by Kang et al., and these methanogenic bacteria could also be the target of future DIET research [56];

2. Using molecular biology and cryoelectron microscopy to clarify how the electron transport chain between electron donor microorganisms and electron acceptor microorganisms completing DIET transfers electrons. The research on DIET is still in the early stage. Although a large number of studies in recent years have proven that CH4 production can be enhanced by DIET, the reported evidence is indirect, and direct evidence of DIET needs to be collected. The cryo-electron microscopy technique has higher resolution, and the microorganisms are better maintained in their original state when frozen at low temperature than when dried, resulting in more objective and direct observations. The filamentous protein appendages known as "microbial nanowires" have been found to be composed not of pili but of the cytochromes *OmcS* and *OmcZ*;

3. How DIETs can adapt to extreme weather at very high or low temperatures without compromising their electron transfer efficiency. In recent years, global environmental degradation has led to global warming, ozone layer depletion, acid rain, freshwater crisis, energy shortage, sharp decrease in forest resources, land desertification, accelerated species extinction, garbage disaster, toxic chemical pollution, and many other aspects of environmental problems. The study of IET is of great importance for biogeochemical cycles, such as carbon cycle, nitrogen cycle, methane production, and greenhouse gas emissions. A future research direction could involve using the DIET process to adapt to extreme weather, such as very high or very low temperature, without affecting its electron transport efficiency. This could also examine microorganisms that produce electricity and are resistant to high or low temperatures.

**Author Contributions:** K.S. selected the topic; collected and sorted data; formulated or evolved overarching research goals and aims. L.L. conducted a research and investigation process, or data/evidence collection. Q.W. designed the research methodology. R.C. analyzed the data. All authors have read and agreed to the published version of the manuscript.

**Funding:** This work was supported by Sichuan Science and Technology Program (2021YFS0284).

**Institutional Review Board Statement:** Not applicable.

**Informed Consent Statement:** Not applicable.

**Data Availability Statement:** Not applicable.

**Conflicts of Interest:** The authors declare no conflict of interest.

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
