# Peer review of "A Review on the Interspecies Electron Transfer of Methane Production in Anaerobic Digestion System"

_fermentation, doi:10.3390/fermentation9050467_

Round 1

Reviewer 1 Report

Please check the sentences between Line 162 to 170! In addition, the content in this part doesn’t seem relevant to the sub-title. Please elaborate.

2011 comes immediately after 2005? Please avoid such sentences!

Line 180 “When the distance between two microorganisms is close enough”!!

What do you mean by distance? How close? And the distance could be measured and evaluated for a higher DIET process?

It is assumed the authors have pilled up certain information without understanding them?

What was the mechanism behind increasing in biogas production after the addition of zero-valent iron (ZVI) ? Such information is indeed, needs to discuss in detail

The authors have added a few of information carbon-based materials impacting the DIET under the sub-section as “Effect of DIET and MIET on methanogenic performance” which is not relevant to this section!

Author Response

Respond to Reviewer Comments

Manuscript ID: fermentation-2337121

Title: A review on the interspecies electron transfer of methane production in anaerobic digestion system

Journal: Fermentation

Dear Reviewer:

Thanks for your helpful comments on our manuscript “A review on the interspecies electron transfer of methane production in anaerobic digestion system”, which are very important for improvement of our manuscript.

We extensively revised the manuscript and highlighted the changes with yellow color in the manuscript. We also replied the comments with a point-by-point response as described below.

Q1: Please check the sentences between Line 162 to 170! In addition, the content in this part doesn’t seem relevant to the sub-title. Please elaborate.

Response: Thank you very much for your comments. Based on your suggestion, we revised the sentences between Line 162 to 170. And revise this section to highlight the key points of this paragraph according to the sub-title. This quote refers to a 2010 article by Summers et al. in Science titled “Direct exchange of electrons within aggregates of an evolved syntrophic co-culture of anaerobic bacteria”. In their study of co-culture systems of G. metallireducens and G. sulfurreducens, they found that the G. sulfurreducens hydrogenase knockout strain was able to co-culture with G. metallireducens even though it was unable to utilize H2. However, when the Multi-haem cytochrome genes, omcS, and the gene related to bacteriophage pili synthesis, pilA, were knocked out in G. sulfurreducens, the growth of the bacterial in the co-culture system was found to be inhibited. This experimental result indicates that electron transfer in the co-culture system is carried out through the conductive bacterial hairs, cytochrome c-linked DIET. This experimental result indicates that electron transfer in the co-culture system is carried out through the conductive bacterial pili, cytochrome c-linked DIET.

Modified in lines 165-172.

Q2: 2011 comes immediately after 2005? Please avoid such sentences!

Response: Thank you very much for your comment. All the sentences in the text such as these have been revised.

Modified in lines 162-172.

Q3: Line 180 “When the distance between two microorganisms is close enough”!! What do you mean by distance? How close? And the distance could be measured and evaluated for a higher DIET process? It is assumed the authors have pilled up certain information without understanding them?

Response: Thank you very much for the comment. The text has been revised in the corresponding place.

Modified in lines 177-178.

Q4: What was the mechanism behind increasing in biogas production after the addition of zero-valent iron (ZVI)? Such information is indeed, needs to discuss in detail

Response: Thank you very much for the comment. The corresponding mechanism for the increase in biogas production after addition of zero-valent iron (ZVI) has been discussed in detail and added in the paper.

Modified in lines 248-258.

Q5: The authors have added a few of information carbon-based materials impacting the DIET under the sub-section as “Effect of DIET and MIET on methanogenic performance” which is not relevant to this section!

Response: Thank you very much for the comment. We have checked the full text and corrected typo errors. In this section, the effects of MIET and DIET on the methanogenic performance of anaerobic digestion systems are presented separately. The interspecies electron transfer by MIET is often limited by electron carriers and vulnerable to external environmental influences; however, in DIET-mediated methanogenesis, electrons are not transferred by diffusive electron carriers, but by their own conductive cellular structures or by external conductive materials for direct electron transfer. The external conductive materials are discussed through the external iron-based materials and the external carbon-based materials, respectively, which are relevant to this section.

Reviewer 2 Report

The work titled "A review on the interspecies electron transfer of methane production in anaerobic digestion system" is clear and well-organized. However, before the paper's publication, the authors should better highlight the goal of the paper: why does their review paper have a significant impact on the scientific community and are worthy of being published?

The authors have to improve the conclusions by highlighting the benefits of the review knowledge concerning not only the scientific point of view but also the technical point of view.

Author Response

Respond to Reviewer Comments

Manuscript ID: fermentation-2337121

Title: A review on the interspecies electron transfer of methane production in anaerobic digestion system

Journal: Fermentation

Dear Reviewer:

Thanks for your helpful comments on our manuscript “A review on the interspecies electron transfer of methane production in anaerobic digestion system”, which are very important for improvement of our manuscript.

We extensively revised the manuscript and highlighted the changes with red color in the manuscript. We also replied the comments with a point-by-point response as described below.

Q1: The work titled "A review on the interspecies electron transfer of methane production in anaerobic digestion system" is clear and well-organized. However, before the paper's publication, the authors should better highlight the goal of the paper: why does their review paper have a significant impact on the scientific community and are worthy of being published?

Response: Thank you very much for your comments. We have revised the goal of the paper in the introduction section.

Modified in lines 87-96.

Q2: The authors have to improve the conclusions by highlighting the benefits of the review knowledge concerning not only the scientific point of view but also the technical point of view.

Response: Thank you very much for your comment. We have added in the conclusion section the relevant the technical point of view.

Modified in lines 510-517 and 526-528.

Reviewer 3 Report

Overall, the paper is well constructed and provides comprehensive insights in this field of research by synthesizes the literature.

Major comments that need to be addressed:

I am wondering why abbreviation for  Indirect Interspecies Electron Transfer is MIET, not IIET like other researchers used.

I see equation (2) C6H12O6+2H2O2CH3COOH+2CO2+4H2 also published by a few researchers, can authors explain to let me understand why in this equation, O is not balanced: 8 O on the left, 6 O on the right of the equation.

Table 1 and Table 2 must be mentioned in the main text rather than just appear in the paper because readers need understand the connection between the tables with the rest of the text.

Please check the whole paper and write concisely. Avoid of expressing same idea twice in one sentence or in one paragraph. E.g.

1)      Line 17: remove “’…involved in MIET and DIET”, it is already mentioned “…two approaches of DIET and MIET…”

2)      in Line 16. E.g. Line 37-45:” The flow chart of anaerobic digestion is shown in Figure 1…. The anaerobic methanogenesis process is shown in Figure 1.”

3)      Line 338 and 339 “In DIET, the electron donor microorganisms  are G. metallireducens, Desulfovibrio, etc.” vs. Line 343-345. “Donor microorganisms include G.mentallireducens GS-15 [18]G.sulfurreducens PCAG.hy- drogenophilus [78]Anaerobic methanotrophic [37, 79] etc.”

 The conclusion, line 513-519. Authors raised a follow-up research on “A more detailed quantification of the electron transfer efficiency of DIET using nano secondary ion mass spectrometry analysis of single cell metabolic activity:...” But neither quantification of the electron transfer efficiency nor any existing analysis methods are reviewed in this paper, this suggestion is not connected to the main content of the paper, some background may be needed.  

Minor revisions:

Please define the abbreviations IHT and IFT when they first appear in the text, ie. Line 86: “… an effective alternative to IHT/IFT”.

 A proof-reading is requested for gramma and typo, E.g.:

Line 306-308, gramma mistake, please modify the sentence “Extensive studies from 2012….processes [62]”.

 Line 312-314, modify the sentence “…Kapp of the Fe3O4 added reactor..” to make it clear.

 Line 345 and line 348, checking sentences, remove unnecessary character , using and.

 Please rewrite the sentences in Line 436-437 and Line 442-446. The similarity check with original reference is too high for these two citations. 

 Line 536-538. Please modify the sentence to make it clear.

Line 464 “…by domestic and foreign researchers…” This journal is an international journal and this is a review paper based on global research findings, there should no “domestic and foreign” for this paper’s content.

 Authors may consider to add some recent references. For example:

 Wang, W.; Lee, D-J. Direct interspecies electron transfer mechanism in enhanced methanogenesis: A mini-review. Bioresource Technology. 2021, 330, 124980 https://doi.org/10.1016/j.biortech.2021.124980

Nguyen, L. N.; Vu, M. T.; Abu Hasan Johir, Md.; et al. Promotion of direct interspecies electron transfer and potential impact of conductive materials in anaerobic digestion and its downstream processing - a critical review. Bioresource Technology. 2021, 341, 125847 https://doi.org/10.1016/j.biortech.2021.125847

Author Response

Respond to Reviewer Comments

Manuscript ID: fermentation-2337121

Title: A review on the interspecies electron transfer of methane production in anaerobic digestion system

Journal: Fermentation

Dear Reviewer:

Thanks for your helpful comments on our manuscript “A review on the interspecies electron transfer of methane production in anaerobic digestion system”, which are very important for improvement of our manuscript.

We extensively revised the manuscript and highlighted the changes with blue color in the manuscript. We also replied the comments with a point-by-point response as described below.

Reviewer #3

Overall, the paper is well constructed and provides comprehensive insights in this field of research by synthesizes the literature.

                                                                                               —— Reviewer 3.

Q1: I am wondering why abbreviation for Indirect Interspecies Electron Transfer is MIET, not IIET like other researchers used.

Response: Thank you very much for your comments. In fact, some researchers use IIET (Interspecies Electron Transfer) and some use MIET (mediated interspecies electron transfer), and the full names and abbreviations have been changed accordingly in the text.

Modified in lines 12.

Q2: I see equation (2) C6H12O6+2H2O→2CH3COOH+2CO2+4H2 also published by a few researchers, can authors explain to let me understand why in this equation, O is not balanced: 8 O on the left, 6 O on the right of the equation.

Response: Thank you very much for your comment. The equation was not leveled due to my negligence. The correct equation has now been modified to the corresponding position in the text.

Modified in lines 51.

Q3: Table 1 and Table 2 must be mentioned in the main text rather than just appear in the paper because readers need understand the connection between the tables with the rest of the text.

Response: Thank you very much for the comment. Tables 1 and 2 have been mentioned in the text at the corresponding places.

Modified in lines 225-226 and 326.

Q4: Please check the whole paper and write concisely. Avoid of expressing same idea twice in one sentence or in one paragraph. E.g.

1) Line 17: remove “’…involved in MIET and DIET”, it is already mentioned “…two approaches of DIET and MIET…”

2) in Line 16. E.g. Line 37-45:” The flow chart of anaerobic digestion is shown in Figure 1…. The anaerobic methanogenesis process is shown in Figure 1.”

3) Line 338 and 339 “In DIET, the electron donor microorganisms  are G. metallireducens, Desulfovibrio, etc.” vs. Line 343-345. “Donor microorganisms include G.mentallireducens GS-15 [18]G.sulfurreducens PCAG.hy- drogenophilus [78]Anaerobic methanotrophic [37, 79] etc.”

Response: Thank you very much for the comment. Sentences with duplicate references have been removed.

Modified in lines 17, 37, 341-342.

Q5: The conclusion, line 513-519. Authors raised a follow-up research on “A more detailed quantification of the electron transfer efficiency of DIET using nano secondary ion mass spectrometry analysis of single cell metabolic activity:...” But neither quantification of the electron transfer efficiency nor any existing analysis methods are reviewed in this paper, this suggestion is not connected to the main content of the paper, some background may be needed. 

Response: Thank you very much for the comment. The part not mentioned in the text has been removed.

Q6: Please define the abbreviations IHT and IFT when they first appear in the text, ie. Line 86: “… an effective alternative to IHT/IFT”.

Response: Thank you very much for the comment. The full name of the abbreviation has been added to the corresponding position in the text.

Modified in lines 85-86.

Q7: A proof-reading is requested for gramma and typo, E.g.: Line 306-308, gramma mistake, please modify the sentence “Extensive studies from 2012….processes [62]”.

Response: Thank you very much for the comment. The grammar of the sentence has been revised.

Modified in lines 305-308.

Q8: Line 312-314, modify the sentence “…Kapp of the Fe3O4 added reactor.” to make it clear.

Response: Thank you very much for the comment. The grammar of the sentence has been revised.

Modified in lines 312-314.

Q9: Line 345 and line 348, checking sentences, remove unnecessary character , using and.

Response: Thank you very much for the comment. The sentence has been revised.

Modified in lines 343-344.

Q10: Please rewrite the sentences in Line 436-437 and Line 442-446. The similarity check with original reference is too high for these two citations.

Response: Thank you very much for the comment. The sentence has been revised.

Modified in lines 435-438 and 442-445.

Q11: Line 536-538. Please modify the sentence to make it clear.

Response: Thank you very much for the comment. The sentence has been revised.

Modified in lines 527-532.

Q12: Line 464 “…by domestic and foreign researchers…” This journal is an international journal and this is a review paper based on global research findings, there should no “domestic and foreign” for this paper’s content.

Response: Thank you very much for the comment. The sentence has been revised.

Modified in lines 464.

Q13: Authors may consider to add some recent references. For example:

Wang, W.; Lee, D-J. Direct interspecies electron transfer mechanism in enhanced methanogenesis: A mini-review. Bioresource Technology. 2021, 330, 124980 https://doi.org/10.1016/j.biortech.2021.124980

Nguyen, L. N.; Vu, M. T.; Abu Hasan Johir, Md.; et al. Promotion of direct interspecies electron transfer and potential impact of conductive materials in anaerobic digestion and its downstream processing - a critical review. Bioresource Technology. 2021, 341, 125847 https://doi.org/10.1016/j.biortech.2021.125847

Response: Thank you very much for the comment. Recommended literature references have been cited in the text.

Modified in lines 247 and 275.

Reviewer 4 Report

This review article provides a comprehensive overview of Interspecies Electron Transfer (IET) in anaerobic digestion systems. The authors have done an excellent job of summarizing the recent research progress on DIET and MIET mechanisms, methane production, thermodynamics, and kinetics. The potential applications of IET in sustainable management of high concentration organic wastewater and fiveare also discussed in detail and ffive following areas of follow-up research. The article is well-structured, logically organized, and clearly written. However, there are a few areas where the authors could improve the article. 

First of all, the discussion part can expand or summarize more practical issues related to IET implementation and future research directions. Listing one or two  practical application studies will be more helpful for readers to understand

Secondly, ome figures or diagrams could be added to help illustrate the concepts discussed in the article. It will be very helpful if the author can add an auxiliary explanatory picture or a simple & vivid description picture to describe this chapter in point 4.

Overall, this review article is an informative review article for researchers and practitioners interested in IET and its potential application in anaerobic digestion systems.

Author Response

Respond to Reviewer Comments

Manuscript ID: fermentation-2337121

Title: A review on the interspecies electron transfer of methane production in anaerobic digestion system

Journal: Fermentation

Dear Reviewer:

Thanks for your helpful comments on our manuscript “A review on the interspecies electron transfer of methane production in anaerobic digestion system”, which are very important for improvement of our manuscript.

We extensively revised the manuscript and highlighted the changes with green color in the manuscript. We also replied the comments with a point-by-point response as described below.

Reviewer #4

This review article provides a comprehensive overview of Interspecies Electron Transfer (IET) in anaerobic digestion systems. The authors have done an excellent job of summarizing the recent research progress on DIET and MIET mechanisms, methane production, thermodynamics, and kinetics. The potential applications of IET in sustainable management of high concentration organic wastewater and five are also discussed in detail and five following areas of follow-up research. The article is well-structured, logically organized, and clearly written. However, there are a few areas where the authors could improve the article.

                                                                                                   —— Reviewer 4.

Q1: First of all, the discussion part can expand or summarize more practical issues related to IET implementation and future research directions. Listing one or two practical application studies will be more helpful for readers to understand.

Response: Thank you very much for your comments. Each point of research direction in the discussion section is supplemented by a list of corresponding practical application studies.

Modified in lines 510-517.

Q2: Secondly, some figures or diagrams could be added to help illustrate the concepts discussed in the article. It will be very helpful if the author can add an auxiliary explanatory picture or a simple & vivid description picture to describe this chapter in point 4.

Response: Thank you very much for your comment. The diagram(Fig.4) corresponding to the point 4 has been added in the text.

Modified in lines 494.